# First Report of the Genus *Quinquelaophonte* Wells, Hicks and Coull, 1982 (Copepoda: Harpacticoida: Laophontidae) from China, with Description of a New Species [note 1]

**DOI:** 10.3390/biology14030271

**Published:** 2025-03-06

**Authors:** Zhengcun Hou, Qi Kou, Lin Ma

**Affiliations:** 1Qingdao Key Laboratory of Marine Biodiversity and Conservation, Department of Marine Organism Taxonomy and Phylogeny, Institute of Oceanology, Chinese Academy of Sciences, Qingdao 266071, China; houzhengcun@qdio.ac.cn (Z.H.); kouqi@qdio.ac.cn (Q.K.); 2University of Chinese Academy of Sciences, Beijing 100049, China; 3Laboratory for Marine Biology and Biotechnology, Qingdao Marine Science and Technology Center, Qingdao 266237, China

**Keywords:** meiofauna, benthic copepods, integrative taxonomy, morphology, molecular phylogeny

## Abstract

*Quinquelaophonte* species are distributed all around the world and inhabit a variety of environments. Some species are sensitive to pollutants and can serve as an effective species for assessing estuarine health. Laophontidae is one of the most speciose families of Harpacticoida, while the diversity of this family is still rarely studied in China. None of *Quinquelaophonte* has been reported in China. During the investigation of meiofaunal diversity in the intertidal zones, a new species and a new record of the genus *Quinquelaophonte* were discovered based on phylogenetic analyses and morphological comparisons. They are described and illustrated in detail. This work provided a better understanding of the biodiversity of the genus.

## 1. Introduction

The genus *Quinquelaophonte* Wells, Hicks & Coull, 1982 belonging to the harpacticoid family Laophontidae T. Scott, 1905 was initially proposed to accommodate the “*quinquespinosa*” species group previously placed in the genus *Heterolaophonte* Lang, 1948 [1]. This genus is characterized by female antennule with fewer than seven segments, caudal rami with a single well-developed terminal seta, and considerable modification of exopods in male swimming legs 2–4 [1].

Members of *Quinquelaophonte* are widely distributed across the globe and inhabit a variety of environments, such as silt, intertidal mud and gravel [1,2,3,4,5,6,7,8,9,10,11,12,13,14]. Certain species have been confirmed to be sensitive to pollutants and can serve as effective species for assessing estuarine health [15,16]. At present, 14 species have been found worldwide [3,4], with four species reported from the northwest Pacific: *Q. bunakenensis* Mielke, 1997 from Sulawesi, Indonesia, *Q. koreana* Lee, 2003, *Q. enormis* Kim, Nam and Lee, 2020, and *Q. sominer* Kim and Lee, 2023, from Korea.

As one of the most speciose families of Harpacticoida Sars G.O., 1903, Laophontidae comprises more than 327 species and subspecies [17]. However, the diversity of this family is still rarely studied in China. So far, only 18 species of eight genera have been reported from Chinese waters [18,19,20], with most recorded from the South China Sea. Prior to the present studies, no species of this family had been reported from the Yellow Sea.

During recent meiofaunal diversity investigations in Shandong, China, samples of sand-dwelling copepods were collected from the intertidal zones of No. 2 Bathing Beach in Qingdao and Yangma Island in Yantai. In the laboratory, these copepod samples were examined and identified to belong to the genus *Quinquelaophonte*. To investigate the relationship between our specimens and those closely related species, we conducted an interspecific genetic distances analysis based on the mitochondrial cytochrome c oxidase subunit I (COI) gene sequences. Additionally, a phylogenetic tree based on COI and 18S rRNA gene sequences was constructed to infer the position of the new species within Laophontidae. Integrative taxonomic analysis results lead to the conclusion that two species of *Quinquelaophonte* were first reported from China, of which one is new and described herein.

## 2. Materials and Methods

### 2.1. Sampling and Sorting

Specimens were collected from the gravelly sandy intertidal zones of Yangma Island (37°28′31″ N, 121°37′23″ E) in Yantai in May 2021 and from the No. 2 Bathing Beach (36°2′54″ N, 120°20′28″ E) in Qingdao in August 2023 and May 2024. Meiobenthos were extracted from the sediments utilizing a 42 μm sieve and washed three times with fresh water in the laboratory. Harpacticoids were isolated from the extracted organisms using a light microscope SMZ1270 (Nikon, Tokyo, Japan). The isolated harpacticoid specimens were preserved in absolute ethanol and stored at –20 °C before DNA extraction.

### 2.2. DNA Extraction, Amplification and Sequencing

Genomic DNA of three individuals (one male and one female of *Quinquelaophonte xinzhengi* sp. nov. and one female of *Q*. *enormis* Kim, Nam and Lee, 2020) was extracted prior to further morphological examinations. Non-destructive DNA extraction of the whole specimen was performed. COI and 18S rRNA genes were amplified and sequenced for phylogeny inference, while the COI sequences were also used for genetic distance comparisons. The primer pairs of 18S-A/B [21] and LCO1490/HCO2198 [22] were used for gene fragment amplification and sequencing. Other molecular experimental procedures employed in this study followed Wu et al. [23]. The purified PCR products were sent to Tsingke Biotech Co., Ltd. (Beijing, China) for bidirectional Sanger sequencing. The forward and reverse sequence fragments were assembled using CONTIG EXPRESS (a component of Vector NTI Suite 6.0, Life Technologies, Carlsbad, CA, USA). Then, the sequences were subjected to BLAST analysis in the NCBI database to ensure that they were not contaminated. NGS sequencing data of *Platychelipus littoralis* Brady, 1880, was downloaded from the NCBI Sequence Read Archive (SRA) database. Contigs containing mitochondrial and ribosomal genes were assembled using MegaHit 1.2.9 [24]. Sequences of COI and 18S genes were identified using BLAST+ 2.12.0 [25] and extracted respectively by mapping them to corresponding reference sequences (sequences from closely related species) using Unipro UGENE 42.0 [26]. Comparative sequences from GenBank are listed in Table 1.

### 2.3. Sequence Alignment and Phylogenetic Analysis

The homologous sequences were aligned with the MAFFT version 7 webserver [27] using the default parameters and manually trimmed to the same length. The Kimura’s 2-parameter (K2P) genetic distances of COI sequences between the new species and three congeners whose sequences were available were calculated using MEGA 6.06 [28]. The highly divergent aligned regions in the 18S dataset were removed using GBlocks 0.91b [29] (GBlocks parameters: minimum length of a block = 5; allowed gap positions = with half). The trimmed alignments were then concatenated into a single dataset consisting of COI and 18S gene sequences using Sequence Matrix 1.8 [30].

Phylogenetic trees were constructed based on the concatenated dataset using both maximum likelihood (ML) and Bayesian inference (BI) methods. The concatenated dataset was partitioned by gene and also by codon position for COI alignment. The best-fit nucleotide substitution models and optimal partition schemes were selected using ModelFinder [31] and implemented in IQ-TREE 2.2.0.3 [32]. The ML analysis was conducted using IQ-TREE 2, and the node supports were evaluated by performing an SH-like approximate likelihood ratio test (aLRT), as well as an ultrafast bootstrap (UFBoot) with 10,000 replicates [33]. The BI tree was reconstructed using MrBayes 3.2.6 [34]. Two independent runs were performed with four Markov Chains for 10,000,000 generations, with sampling every 1000 generations. After the first 25% (2500) trees were discarded as burn-in, the remaining trees were used to construct the 50% majority rule consensus tree and to estimate the posterior probabilities (PPs). The effective sample size (ESS) values for all sampled parameters were diagnosed using Tracer 1.7.1 [35] to make sure that convergence was reached. The phylogenetic trees were visualized using FigTree 1.4.3 [36]. Finally, all the new sequences were submitted to the GenBank database.

### 2.4. Morphological Identification

After DNA extraction, the exoskeleton of each specimen, along with other specimens preserved in absolute ethanol, was dissected and examined under a stereomicroscope SMZ1270 (Nikon, Tokyo, Japan). Prior to dissection, the body length was measured. Specimens were dissected in lactic acid using tungsten needles and mounted on slides, which were subsequently covered with a coverslip and sealed with nail polish for detailed observations. Observations and pencil illustrations of the whole specimen and dissected appendages were performed using a differential interference contrast microscope Eclipse Ni (Nikon, Tokyo, Japan) equipped with a camera lucida. The habitus was illustrated at 400× magnification, while the body parts were illustrated at 1000× magnification utilizing an oil immersion lens.

The descriptive terminology adheres to the conventions established by Huys et al. [37]. The abbreviations utilized in the text and figures are as follows: A1, antennule; aes, aesthetasc; A2, antenna; Mxp, maxilliped; P1–P6, first through sixth thoracic legs; and exp (enp) -1 (-2, -3), representing the proximal (middle, distal) segment of a ramus. Body length was measured from the anterior margin of the rostrum to the posterior margin of the caudal rami. The type materials are housed at the Marine Biological Museum of the Chinese Academy of Sciences in Qingdao, China (MBM).

## 3. Results

### 3.1. Systematics

Order Harpacticoida Sars, 1903

Family Laophontidae T. Scott, 1905

Genus *Quinquelaophonte* Wells, Hicks and Coull, 1982

Type species. *Quinquelaophonte quinquespinosa* (Sewell, 1924)


***Quinquelaophonte xinzhengi* sp. nov. (Figure 1, Figure 2, Figure 3, Figure 4, Figure 5, Figure 6, Figure 7 and Figure 8)**



**urn:lsid:zoobank.org:act:27373718-2780-4DAF-A238-623517C5B6C6**


**Type locality.** No. 2 Bathing Beach (36°2′54″ N, 120°20′28″ E), Qingdao, China, gravelly sand.

**Material examined.** Holotype: 1♀ for DNA sequencing, dissected on two slides (MBM189284), collected on 24 August 2023; paratypes: 1♀ dissected on three slides (MBM189285), collected on 24 August 2023, 1♀ dissected on four slides (MBM189286), collected on 24 August 2023, 1♀ dissected on three slides (MBM189287), collected on 22 May 2024, 1♂ for DNA sequencing, dissected on three slides (MBM189288), collected on 24 August 2023, 1♂ dissected on three slides (MBM189289), collected on 22 May 2024. Other materials: 1♀ and 2♂ were preserved in 75% ethanol, collected on 22 May 2024. All specimens were collected from the type locality.

**Etymology.** The species is named after Professor Xinzheng Li, principal investigator at the Institute of Oceanology, Chinese Academy of Sciences (IOCAS), for his great contributions to the marine invertebrate diversity research in China.

**Diagnosis.** Body slender, slightly depressed, without hyaline frills on dorsal view. Each somite is well demarcated from others, without distinct demarcation between prosome and urosome. Abdominal somites with row of spinules posteriorly on ventral surface. Caudal rami about 3.2 times as long as maximum width, origin of setae I adjacent to or seperated from setae II and III. Antennule six-segmented in female and seven-segmented in male. Antenna exopod with three setae. Gnathobase of mandible with one seta on distal margin. Syncoxa of maxilliped with two setae at middle of terminal margin. Female P5 baseoendopod bearing five setae and exopod with six setae. Male P5 reduced to five unequal setae. Setal formulae of the swimming legs P2–P4 as follows:

ExpEnpP10.0230.020P20.1.1230.120P30.1.2230.221P40.1230.120


**Description of females.**


**Habitus** (Figure 1A) Total body length ranging from 503 to 840 µm (n = 5, mean = 625 µm, measured from anterior tip of rostrum to posterior margin of caudal rami in dorsal view); largest width from 97 to 165 µm (n = 5, mean = 113 µm, measured at posterior margin of cephalic shield). Body slender, nine-segmented, slightly depressed, with sensilla on all somites except penultimate somite, entire surface without hyaline frills on dorsal view. Each somite is well demarcated from others, without distinct demarcation between prosome and urosome.

**Prosome** (Figure 1A) four-segmented, including cephalothorax (where cephalosome fused with first pedigerous somite) and three free pedigerous somites. All succeeding prosomites combined slightly longer than cephalothorax. Rostrum small, pointed at anterior apex, fused to cephalic shield, with pair of sensilla near anterior margin. Eye not discernible.

**Urosome** (Figure 2A–C) five-segmented, somewhat longer than prosome, consisting of P5 bearing somite, genital double-somite (second urosomite fused to succeeding urosomite), two free abdominal somites, and anal somite. Abdominal somites with row of spinules posteriorly on ventral surface. Genital double-somite longest, completely fused along ventral surface, with deep suture indicating original segmentation between genital somite and third urosomite dorso-laterally and dividing double-somite into equally long halves. Genital field anteriorly with vestigial P6 represented by three setae, genital apertures unobserved, possibly covered by P6. Anal operculum with a row of minute setules terminally, flanked by sensillum on each side.

**Caudal rami** (Figure 2A) elongated, tapering posteriorly, slightly conical, maximum length about 3.2 times as long as maximum width, with seven naked setae on each ramus: origins of setae I–III closely adjacent, arising from posterior half of outer margin, seta I shortest; seta IV–VI located terminally, seta V longest, approximately equal to combined lengths of urosomites 2–5. Seta VII borne on a pedestal at a level similar to origins of setae I–III.

**Antennule** (Figure 1B) six-segmented, gradually tapering forward, with two aesthetascs. First segment with inner seta on anterior corner, with row of spinules near base of inner seta. Second segment slightly longer than width. Third segment about as long as second segment. Fourth segment with terminal aesthetasc fused basally to two closely situated setae. Fifth with one seta. Sixth segment longer than fourth and fifth segments combined, about as long as second and third segments combined, with most setae and terminal aesthetasc. Setal formula: 1–[1], 2–[8], 3–[7], 4–[2 + aes], 5–[1], 6–[10 + aes].

**Antenna** (Figure 1C) four-segmented, including coxa, allobasis, exopod and endopod. Coxa short, without seta and spinule. Allobasis about twice as long as wide, with one tiny abexopodal seta at mid-length and row of spinules nearby. Exopod reduced, with three tiny terminal setae. Endopod one-segmented, broadening distally, about as long as allobasis, with row of setal along innerside and transverse frills on anterior and posterior surfaces near apex; bearing two stout spines near distal part and six terminal setae, of which two bare, three geniculate, and one slender, closely situated to geniculate setae.

**Figure 1 biology-14-00271-f001:**
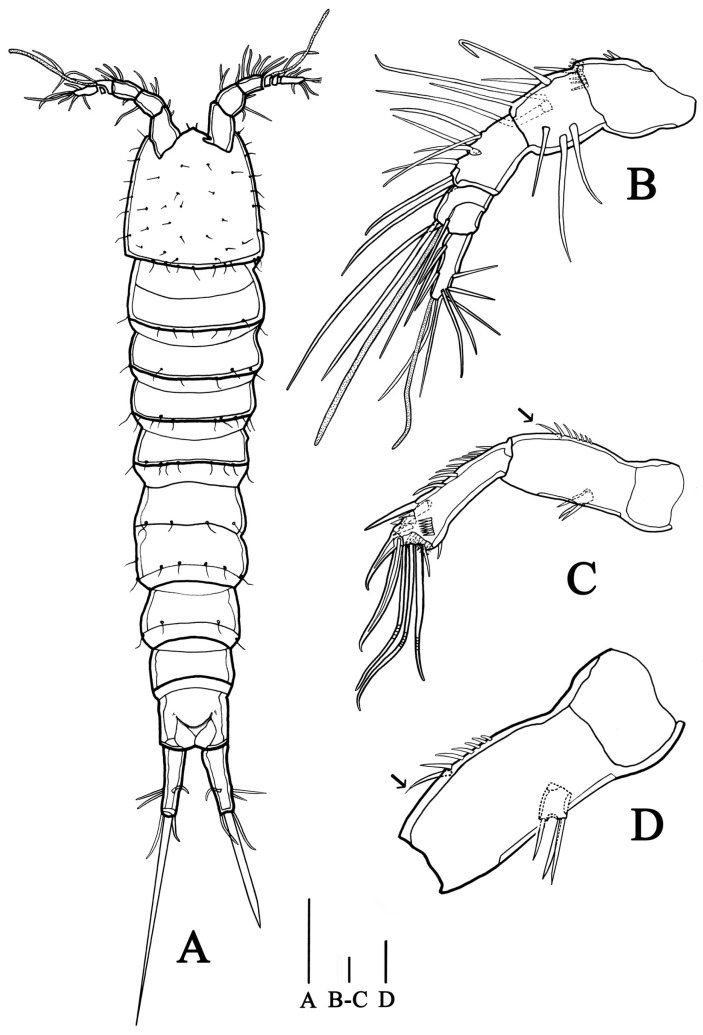
*Quinquelaophonte xinzhengi* sp. nov. female: (**A**) habitus, dorsal (Paratype MBM189286); (**B**) antennule (Paratype MBM189286); (**C**) antenna (Paratype MBM189287); (**D**) allobasis of antenna (Paratype MBM189286). Scale bars: A = 100 μm; B–D = 10 μm.

**Figure 2 biology-14-00271-f002:**
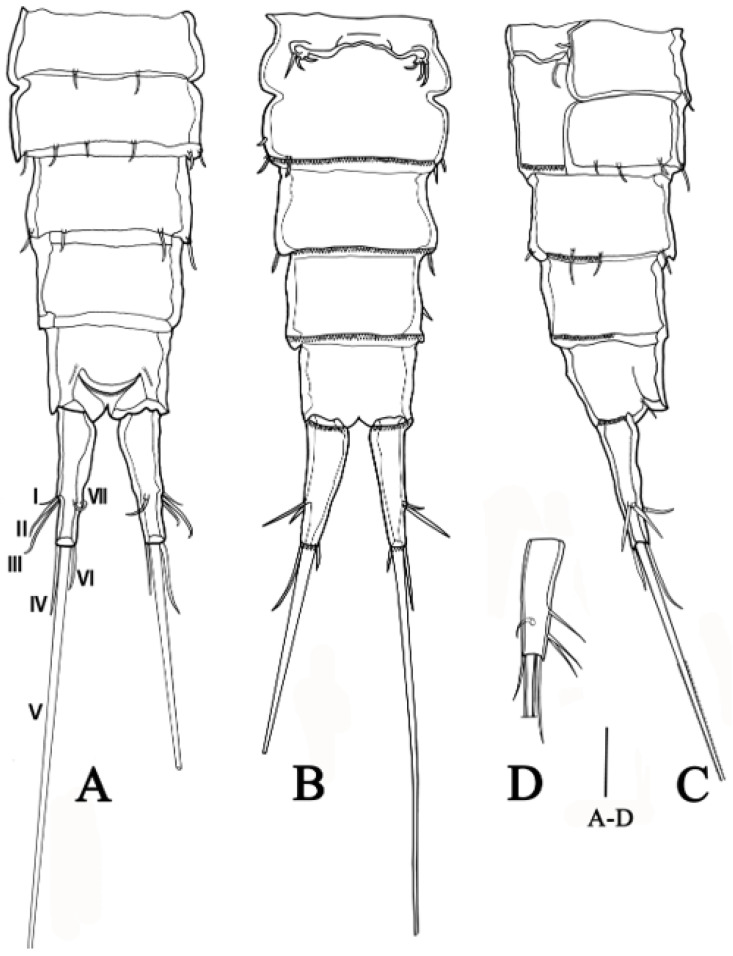
*Quinquelaophonte xinzhengi* sp. nov. female: (**A**) urosome, dorsal (Paratype MBM189286); (**B**) urosome, ventral (Paratype MBM189286); (**C**) urosome, lateral (Paratype MBM189286); (**D**) caudal ramus, ventral (Holotype MBM189284). Scale bar: A–D = 50 μm.

**Mandible** (Figure 3A, B) with well-developed gnathobase, bearing several multicuspidate teeth along distal margin and one long bare seta at dorsal corner. Endopod and exopod fused to basis, beyond recognition. Basis with one terminal seta. Endopod represented by three setae; exopod represented by one naked seta.

**Maxillule** (Figure 3C) five-segmented, including praecoxa, coxa, basis, endopod, and exopod. Praecoxa with one row of spinules on terminal margin, arthrite developed, with six apical spines and one lateral bare seta. Coxa distinct, with cylindrical endite and two long terminal setae. Basis with two setae on endite, one row of spinules on terminal surface. Endopod fused to basis basally, represented by two naked setae. Exopod similar to endopod, also fused to basis, located at middle margin of basis, with one long and one short setae.

**Maxilla** (Figure 3D) three-segmented, including syncoxa, allobasis, and endopod. Syncoxa with two rows of spinules on anterior surface, one row of spinules on posterior surfaces and three closely situated endites; proximal endite with one slender seta; middle endite bearing two strong pectinate spines and one seta, with scattered spinules at base; distal endite with three setae. Allobasis elongated, fused terminally into one strong and pinnate curved claw, bearing three setae. Endopod incorporated into basis, consisting of two naked setae.

**Maxilliped** (Figure 3E) three-segmented, including syncoxa, basis, and endopod. Syncoxa with two setae at middle of terminal margin, with one row of spinules distally. Basis unornamented, about twice as long syncoxa. Endopod one-segmented, tapering to point, with claw and accessory seta at base.

**Figure 3 biology-14-00271-f003:**
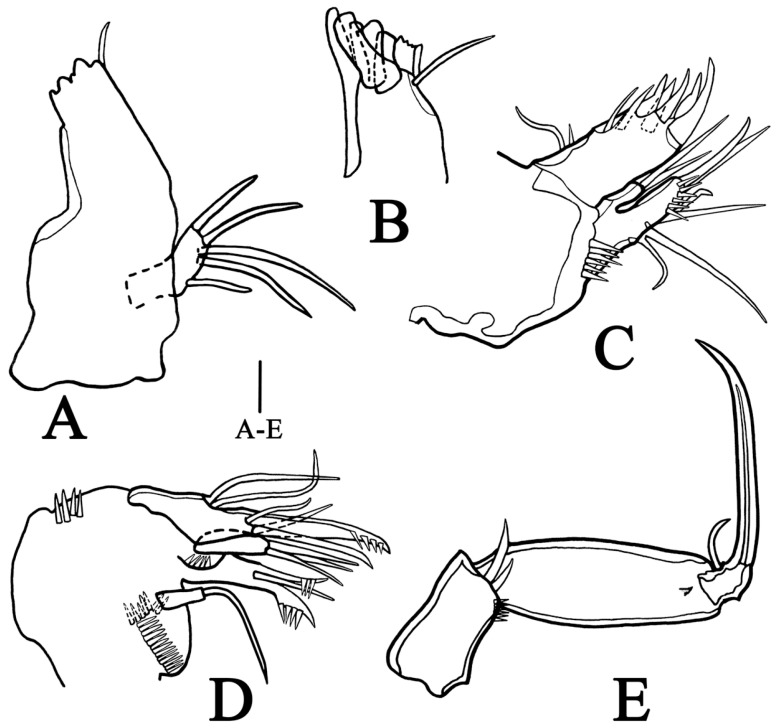
*Quinquelaophonte xinzhengi* sp. nov. female: (**A**) mandible (Paratype MBM189287). (**B**) gnathobase of mandible (Paratype MBM189286). (**C**) maxillule (Paratype MBM189287). (**D**) maxilla (Paratype MBM189285). (**E**) maxilliped (Paratype MBM189285). Scale bar: A–E = 10 μm.

**P1** (Figure 4A) Intercoxal sclerite narrow and naked. Coxa large and robust, with one row of small spinules along middle of outer margin. Basis as long as maximum width, with one strong spine and two rows of scattered small spinules on anterior surface and row of spinules on terminal margin, one pinnate spine on outer side. Exopod divided into two segments, extending to half-length of enp-1; exp-1 with scattered small spinules on outer side and one strong spine near distal margin; exp-2 with row of scattered small spinules and three strong spines on outer side, two geniculate setae at terminal end. Endopod prehensile, divided into two segments; enp-1 with scattered small spinules on upper and middle inner side, spinose projection on anterior surface close to end; enp-2 about one-fourth length of enp-1, terminal margin with a long claw, one setule and one small spinule.

**P2** (Figure 4B) Intercoxal sclerite with two lateral blunt projections. Praecoxa with row of small spinules on distal margin. Coxa covered with row of spinules on anterior surface and outer margin, respectively, scattered spinules along distal margin. Basis narrower than coxa, with one bipinnate spine on outer edge. Exopod longer than endopod, consisting of three segments, tapering distally; exp-1 with one row of spinules on outer edge and middle of anterior surface, respectively, bearing one strong bipinnate spine on outer edge; exp-2 with one strong bipinnate spine and row of spinules on outer margin, one slender seta on inner edge; exp-3 slender, with one inner seta, two plumose and one pinnate terminal setae and two bipinnate outer spines. Endopod two-segmented, all segment with row of setules along inner margin; enp-2 about equal to enp-1, bearing one inner and two terminal plumose setae.

**Figure 4 biology-14-00271-f004:**
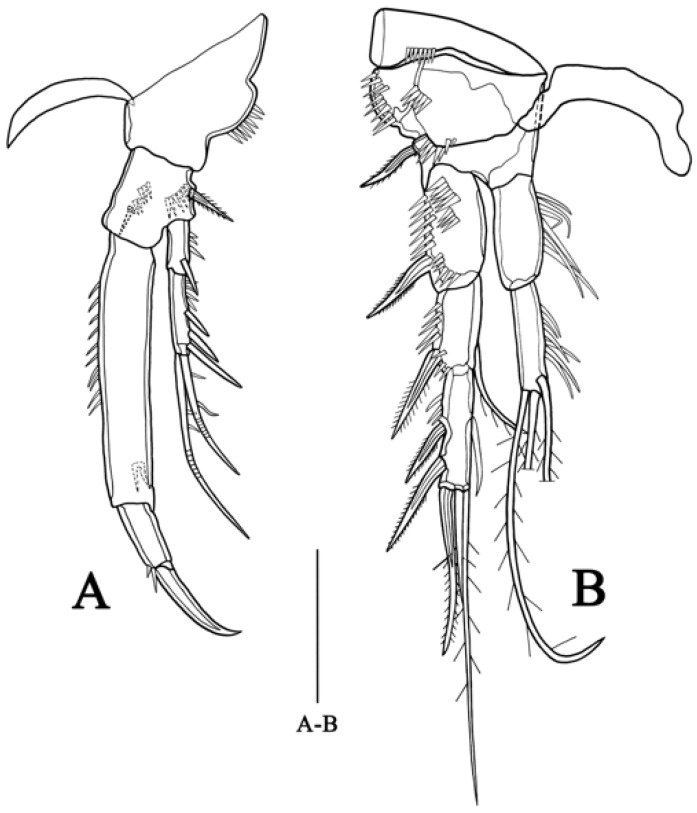
*Quinquelaophonte xinzhengi* sp. nov. female: (**A**) P1, posterior (Holotype MBM189284). (**B**) P2, anterior (Holotype MBM189284). Scale bar: A–B = 50 μm.

**P3** (Figure 5A) Intercoxal sclerite narrow. Praecoxa triangle, with one row of spinules on anterior distal corner. Coxa connected to intercoxal sclerite, with one row of robust spinules on outer side. Basis with row of spinules on outer side, with one naked seta borne on a pedestal. Exopod longer than endopod, all segments with row of spinules on outer margin; exp-1 without inner seta; exp-2 with one inner seta; exp-3 bearing two inner, two distal setae and three outer spines. Endopod two-segmented, enp-1 without inner seta; enp-2 with two inner, two distal and one ouer setae, all setae plumose.

**P4** (Figure 5B) Praecoxa, coxa and basis all with spinular patch around anterior distal corner; basis with one bare issued from one pedestal on outer margin. Exopod three-segmented; exp-1 with scattered small spinules on outer proximal side, and one setae at distal end with one row of spinules around its base; exp-2 with two small spinules and one strong spine on outer margin, one bare seta on inner edge; exp-3 with two small spinules and three strong spines on outer margin, two long setae at tip, and one short seta emerging from inner middle side. Endopod shorter than exopod; enp-1 with one setal on inner margin; enp-2 with one inner plumose and two distal naked setae.

**P5** (Figure 5C) separated, comprising baseoendopod and exopod, with baseoendopod formed by fusion of basis and endopod, without distinct surface sutures marking original segmentation. Baseoendopod with scattered small spinules along outer and distal margin, bearing two bipinnate setae on inner edge, three naked setae on terminal margin. Exopod not fused to basis; with one row of spinules on outer side, scattered small spinules on inner side, and six setae at tip, each arising from one distinct cylindrical process.

**Figure 5 biology-14-00271-f005:**
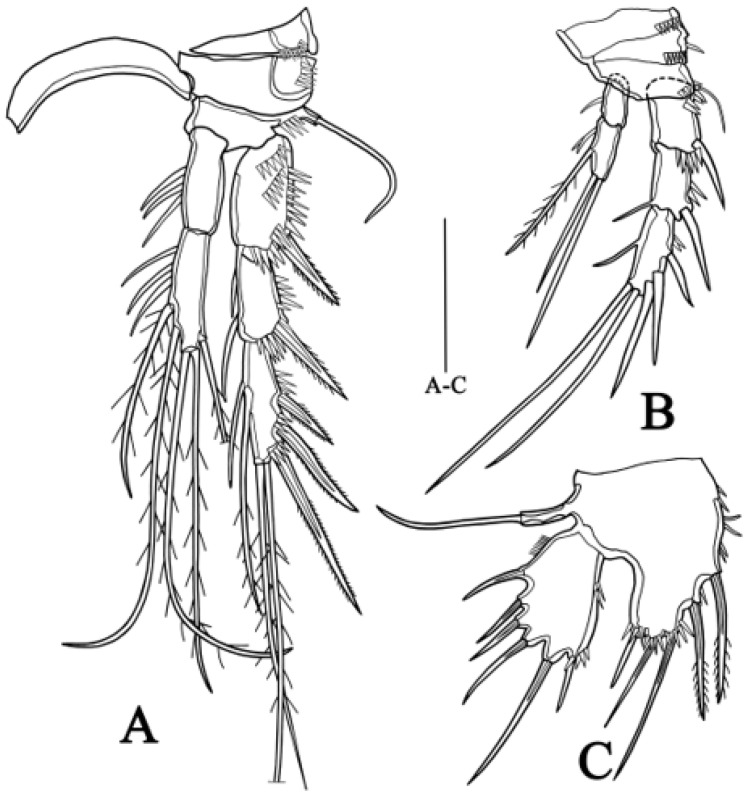
*Quinquelaophonte xinzhengi* sp. nov. female: (**A**) P3, anterior (Holotype MBM189284); (**B**) P4, anterior (Paratype MBM189287); (**C**) P5, anterior (Holotype MBM189284). Scale bar: A–C = 50 μm.

**P6** (Figure 2B) with small protuberance bearing three bare setae, of which outer one longer than two inner ones.


**Description of males.**


The male differs from the female in the following aspects:

**Habitus** (Figure 6A) Total body length ranging from 503 to 782 µm (n = 4, mean = 617 µm, measured from anterior tip of rostrum to posterior margin of caudal rami in dorsal view); Largest width from 105 to 152 µm (n = 4, mean = 125 µm, measured at posterior margin of cephalic shield). Body similar to female, except second and third urosomites not fused. Caudal ramus more elongated.

**Figure 6 biology-14-00271-f006:**
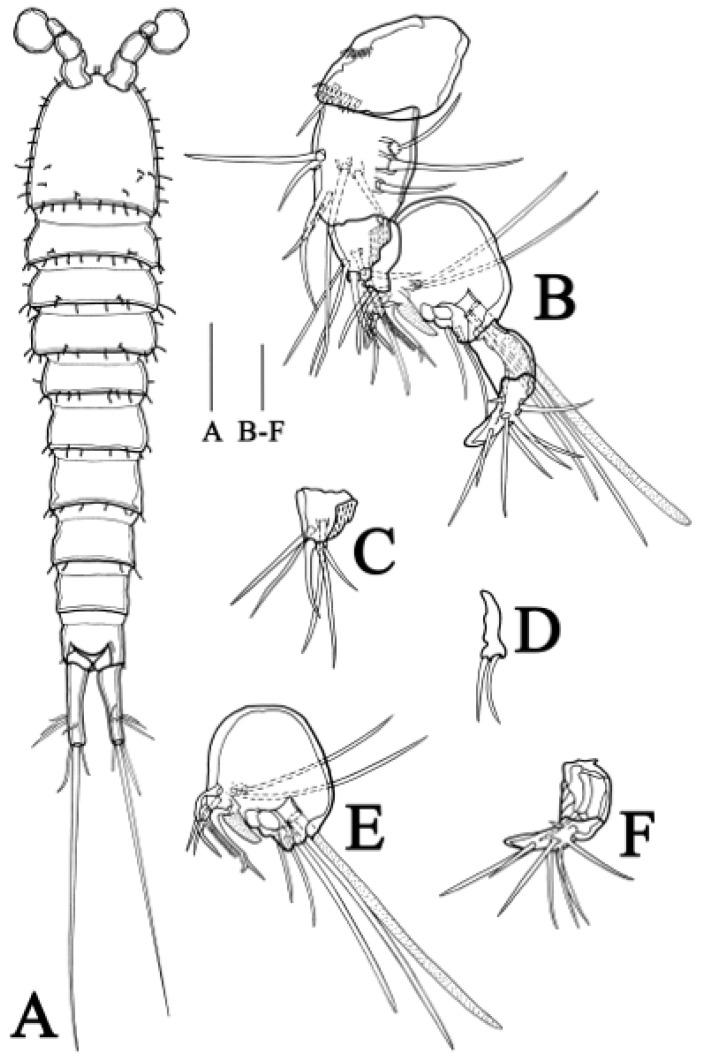
*Quinquelaophonte xinzhengi* sp. nov. male: (**A**) habitus, dorsal (Paratype MBM189288); (**B**) antennule (Paratype MBM189288). (**C**–**F**) third to seven segment of antennule (Paratype MBM189288). Scale bars: A = 100 μm; B–F = 20 μm.

**Urosome** (Figure 7A) genital segment unfused, six-segmented. The penultimate segment without sensory organ. Caudal ramus slightly more elongated than in female, about 4.4 times as long as maximum width, bearing seven setae, distributed in the same pattern as in female.

**Antennule** (Figure 6B–F) subchirocer, seven-segmented. First segment with rows of spinules on distal corner and middle of lateral side, bearing one naked seta on distal corner. Second segment nearly square-shaped, with three setae located on lateral margin, three setae on ventral surface, three setae on outer surface each arising from one pedestal. Third segment triangle, bearing six setae. Fourth segment smallest, with two distal setae. Fifth segment expanded into near-circular shape, with three projections on inner margin’s middle part, including one curved seta, one pectinate seta, two small setae, and one rod-like process on lateral margin; two long setae each issued from one pedestal on ventral surface, four setae and one aesthetasc originated from distal pedestal, aesthetasc fused basally with closest seta. Sixth segment with one distal naked seta. Seventh segment with one spiniform process on distal edge, seven naked setae on outer side and one small one on inner side. Armature formula: 1-[1], 2-[9], 3-[6], 4-[2], 5-[9 + 1 pectinate process + 1 rod-like process + aes], 6-[1], 7-[8 + 1 spiniform process].

**P2** (Figure 7B) Intercoxal sclerite more robust and sturdy than female. Coxa with less ornamentation, only with one small spinule at outer distal corner. Outer bipinnate spine of basis relatively slender and longer than female. Exopod more robust than female; inner seta of exp-2 naked and slender, terminal setae of exp-3 expressed as very robust spines. Endopod more elongated than female; enp-2 with scattered setules on outer side, which absent in female; inner seta bare.

**Figure 7 biology-14-00271-f007:**
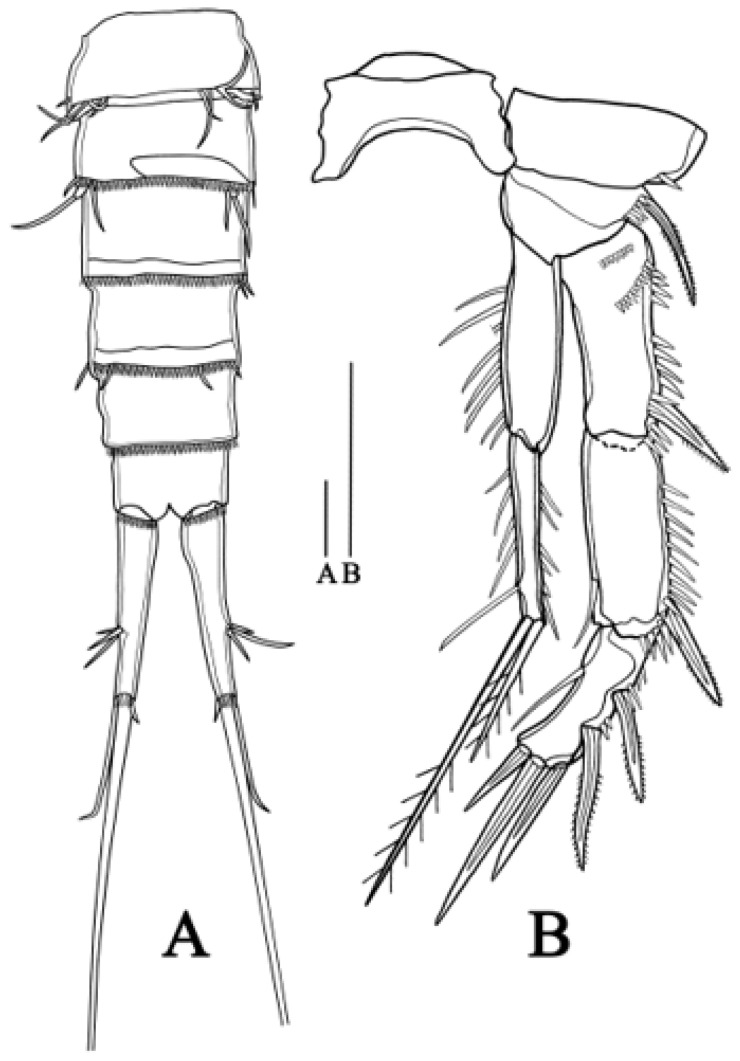
*Quinquelaophonte xinzhengi* sp. nov. male: (**A**) urosome, ventral (Paratype MBM189288); (**B**) P2, anterior (Paratype MBM189288). Scale bar: A–B = 50 μm.

**P3** (Figure 8A) Intercoxal sclerite stouter than female. Praecoxa without ornamentation. Coxa with two rows of spinules on anterior surface. Armature and distribution of basis similar to female. Exopod stronger than that of females; exp-1 with fewer spinules on anterior surface compared with that in females; exp-2 with smaller inner seta than female; exp-3 with one small inner seta, two distal and two outer robust spines. Enp-1 and enp-2 with scattered spinules on outer margins, which absent in female; enp-2 bearing two inner naked setae, two distal plumose setae, and one outer stout spine.

**P4** (Figure 8B) more robust than female. Exp-1 with row of setules along inner margin, row of spinules along outer margin extending to tip, bearing one spine much stouter than female; exp-2 with slender seta on inner side, and distribution of spinules on outer side similar to exp-1; exp-3 shorter and stouter than female, with spinules distributed along outer and terminal edges, bearing one bare inner setae, two distal and three outer stout spines.

**P5** (Figure 8C) reduced to five unequal setae, of which outermost seta issued from one long pedestal, and other four setae arising from small protrusion, innermost one shortest.

**P6** (Figure 7A) Left P6 connected to genital lappet, with two setae, outer one longest and arising from one pedestal. Right P6 simplified to two setae, similar to left.


**Variability**


Five female and four male specimens were examined, including variations in the habitus, mouthparts and setal formulae of the swimming legs. Most morphological features are conservative, except for the body length, location of seta I in the caudal ramus, the ornamentation of maxilliped, and the first swimming legs. The body length of females ranges from 503 to 840 µm, with the largest width ranging from 97 to 165 µm; the males range from 503 to 782 µm, with the largest width ranging from 105 to 152 µm. The origin of seta I is adjacent to setae II and III in the caudal ramus, arising from the posterior half of the outer margin, except for two females with seta I distinctly separated from setae II and III (Figure 2D). The basis of maxilliped has a cuticular bulge subapically in two females, and P1 enp-1 has a spinose projection on the anterior surface close to the end. Setal variations of swimming legs P2–P5 were not found in the examined materials of new species.

**Figure 8 biology-14-00271-f008:**
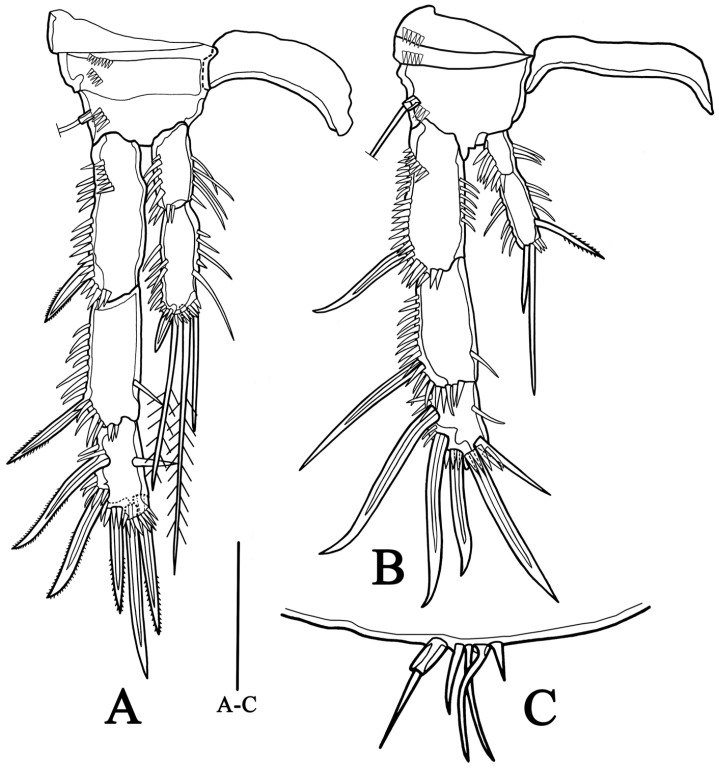
*Quinquelaophonte xinzhengi* sp. nov. male: (**A**) P3, anterior (Paratype MBM189288); (**B**) P4, anterior (Paratype MBM189288); (**C**) P5, anterior (Paratype MBM189288). Scale bar: A–C = 50 μm.


***Quinquelaophonte enormis* Kim, Nam & Lee, 2020 (Figure 9)**


**Type locality.** Gijang, Busan, Korea, east coast of Korea (35°16′3.95″ N; 129°14′ 39.72″ E), sandy shore.

**Material examined.** 3♀ (MBM 189290–189292), each one dissected and mounted on three slides labeled respectively, 1♀ for DNA sequencing (MBM 189290). All specimens collected from Yangma Island (37°28′31″ N, 121°37′23″ E) on 7 May 2021.

**Notes on the description of the female.** The morphology of our specimens is generally in accordance with Kim et al.’s [3] original description, except some minor differences as follows: (1) basis of maxilliped (Figure 9E) with cuticular bulge subapically; (2) P1 (Figure 9A) enp-1 with cuticular bulge subapically, P1 exp-2 with more spinules on outer margin; (3) P2 (Figure 9B) enp-1 without tube pore presented on outer distal corner; (4) P3 bilateral symmetry, exp-3 (Figure 9C) without inner seta; (5) P4 bilateral symmetry, exp-2 (Figure 9D) without inner seta.

**Figure 9 biology-14-00271-f009:**
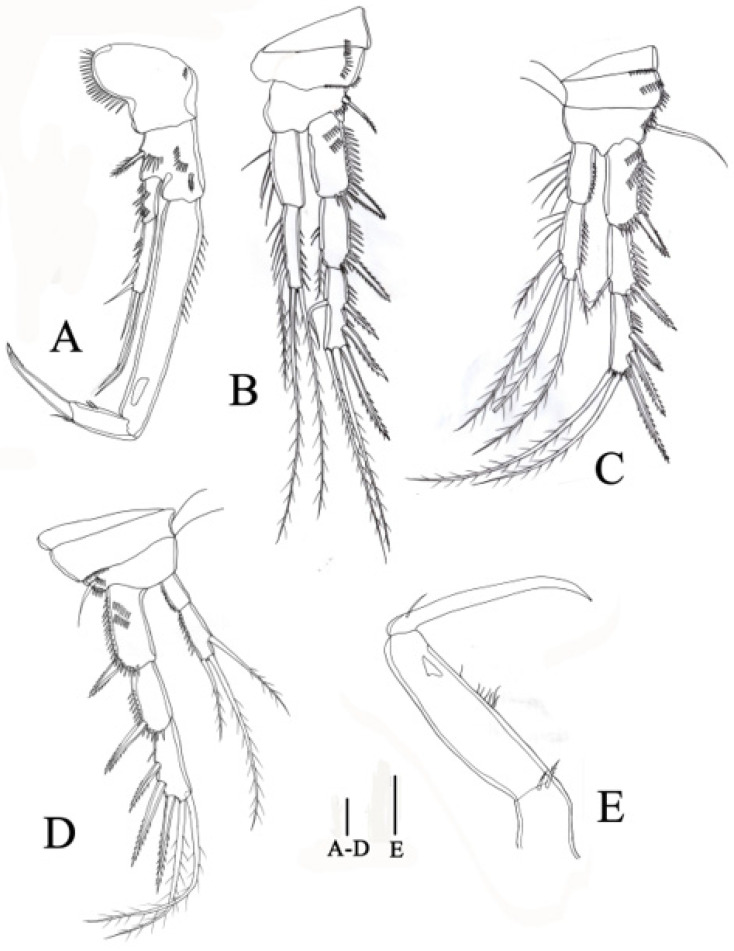
*Quinquelaophonte enormis* Kim, Nam & Lee, 2020. female (MBM 189290): (**A**) P1, anterior; (**B**) P2, anterior; (**C**) P3, anterior; (**D**) P4, anterior; (**E**) maxilliped, anterior. Scale bars: A–E = 10 μm.

### 3.2. Molecular Analyses

The intrageneric COI genetic divergences based on the alignment of 681 bp ranged from 19.1% (between *Q*. *aurantius* Charry, Wells, Smith, Stringer & Tremblay, 2019 and *Q*. *enormis*) to 22.4% (between *Q*. *enormis* and *Q*. *sominer* Kim & Lee, 2023). The low genetic divergences between the Korean individual of *Q. enormis* and the individual from Yantai, China (1.1%) suggested that they are conspecific despite minor morphological variabilities. In addition, the genetic divergences between the new species and three congeneric species all exceed 20% (21.5–22.3%), supporting their separate species status. Since no genetic divergence was observed between the male and female individuals from Qingdao, China, they are confirmed to be the same species, and the morphological differences between them should be regarded as sexual dimorphism (Table 2).

The concatenated dataset consisted of 2351 bp (~93.25% of the original 2521 bp alignment) after the alignments were trimmed with GBlocks (original alignment of 18S = 1861 bp, trimmed alignment of 18S = 1691 bp). The optimal partitioning scheme and the best-fit models for each partitioned subset selected by ModelFinder are listed in Table 3. Both ML and BI analyses strongly supported the monophyly of Laophontidae (aLRT and UFBoot = 100%, PP = 1.00) with Harpacticidae Dana, 1846 as the outgroup (represented by two species of *Tigriopus* Norman, 1869). *Quinquelaophonte* was also suggested to be monophyletic with high support (aLRT and UFBoot ≥ 99.9%, PP = 1.00). In contrast, some internal relationships among genera of Laophontidae were poorly resolved due to the low node support values or inconsistent topologies between ML and BI inferences. Specifically, in the ML tree, *Quinquelaophonte* is sister to the clade comprising *Platychelipus littoralis* Brady, 1880, *Paralaophonte* (*Paralaophonte*) *congenera* (Sars G. O., 1908), *Vostoklaophonte eupenta* Yeom, Nikitin, Ivanenko & Lee, 2018 and *Microchelonia koreensis* (Kim I. H., 1991). However, this clade first clustered with another small clade consisting of *Pseudonychocamptus spinifer* Lang, 1965 and *Laophontina* sp. in the BI tree. The two different groupings were both weakly supported (aLRT = 30, UFBoot = 47%, PP = 0.85) (Figure 10), which should be attributed to the insufficient taxon sampling and limited genetic data.

### 3.3. Morphological Characters Comparison of Quinquelaophonte Species

In order to distinguish the new species from its congeners, a comparison of the most significant morphological characters among all the valid species of *Quinquelaophonte* is provided (Table 4). All morphological character states were collected from original descriptions and reliable redescriptions, except for the new species.

## 4. Discussion

Both morphological comparison and molecular phylogenetic analysis suggested that *Quinquelaophonte xinzhengi* sp. nov. is most closely related to *Q. aurantius* Charry, Wells, Smith, Stringer and Tremblay, 2019. However, the COI genetic distance between the two species reaches to 26.2%, and they can be clearly separated by the following features: (1) *Q. xinzhengi* sp. nov. has one seta on the distal margin of mandible gnathobase, which is absent in *Q. aurantius*; (2) *Q. xinzhengi* sp. nov. bears two inner setae on female P3 exp-3, while *Q. aurantius* only has one.

Regarding the geographic distribution, four other species have been reported from the Northwest Pacific. Among them, *Quinquelaophonte xinzhengi* sp. nov. resembles *Q. enormis* and *Q. sominer* the most. They can be distinguished by the following features: (1) *Q. xinzhengi* sp. nov. has one seta on the distal margin of mandible gnathobase, which is absent in *Q. enormis* and *Q. sominer*; (2) the length ratio of P1 enp-1 is about 5.0 in *Q. xinzhengi* sp. nov., while it is about 6.2 in *Q. enormis* and 6.0 in *Q. sominer*; (3) *Q. xinzhengi* sp. nov. bears two inner setae on female P3 exp-3, while both *Q. enormis* and *Q. sominer* have only one inner seta; (4) *Q. xinzhengi* sp. nov. and *Q. sominer* bear one inner seta on female P4 exp-3, which is absent in *Q. enormis.* In addition, *Q. xinzhengi* sp. nov. can be discriminated with another Northwest Pacific species, *Q. bunakenensis,* by female caudal ramus being 3.2 times as long as the width and P3 enp-2 with five elements (vs. caudal ramus being 2.1 times as long as width, P3 enp-2 with the four elements). *Q. xinzhengi* sp. nov. also differs from *Q. koreana* by female caudal ramus being 3.2 times as long as the width and P4 enp-2 having four elements (vs. caudal ramus being 1.1 times as long as the width and P4 enp-2 with five elements).

In addition to the five species mentioned above, *Quinquelaophonte xinzhengi* sp. nov. can be distinguished from *Q. aestuarii* Sciberras, Bulnes and Cazzaniga, 2014, *Q. varians* Bjornberg, 2010, *Q. prolixasetae* Walker-Smith, 2004, *Q. parasigmoides* (Bozic, 1969) and *Q. longifurcata* (Lang, 1965) by the female P3 exp-3 having two inner setae, whereas these five species lack the inner seta or present only one. *Q. xinzhengi* sp. nov. can be distinguished from *Q. candelabrum* Wells, Hicks and Coull, 1982, by six-segmented antennule and P5 exopod having six setae in the female (vs. five-segmented antennule and P5 exopod with five setae in females). *Q. xinzhengi* sp. nov. can be differentiated from *Q. wellsi* (Hamond, 1973) by the morphology of the seta at the tip of P1 enp-2, which is a minute setule in *Q. xinzhengi* sp. nov. but a long seta in *Q. wellsi*. *Q. xinzhengi* sp. nov. is distinct from two widely distributed species, *Q*. *capillata* (Wilson, 1932) and *Q. quinquespinosa* (Sewell, 1924), in the length ratio of caudal ramus, the morphology of seta at the tip of P1 enp-2 and number of elements in female P2 enp-2, P4 enp-2 and P3-P4 exp-3 (see Table 4).

The morphological and molecular data evaluations validate the new species of *Quinquelaophonte*, as well as expand the distribution of *Q. enormis* from the South Sea of Korea to the North Yellow Sea. For one thing, five species of the genus *Quinquelaophonte* are currently reported from this concentrated area of the Northwest Pacific, suggesting the diversity of this genus and other harpacticoids could be far beyond our expectations. For another, despite high diversity, limited genetic data of benthic copepods are available in public databases (e.g., NCBI, BOLD) for species identification and phylogeny inference, impeding us from understanding the diversity and evolution of this group with great ecological significance. Admittedly, the small size increases the difficulty of acquiring molecular data from benthic copepods. However, with the development of new sequencing techniques combined with data analysis methods, we are confident that more and more molecular data from diverse copepod groups will be obtainable in the near future.

Key to the Northwest Pacific species of *Quinquelaophonte* Wells, Hicks and Coull, 1982 (based on characters of females)
1.Caudal ramus less than 3 times as long as width…………………………………………………………………………………………………….2Caudal ramus at least 3 times as long as width………………………………………………………………………………………………………32.P3 enp-2 with 4 elements, P4 enp-2 with 3 elements………………………………………………………………………..…………*Q. bunakensis*P3 enp-2 with 5 elements, P4 enp-2 with 4 elements………………………………………………………………………………………*Q. koreana*3.P3 exp-3 with 2 inner setae……………………………………………………………………………………………………..…*Q. xinzhengi* sp. nov.P3 exp-3 with 1 inner setae………………………………………………………………………………………………………………………………44.All body somite without hyaline frills, P4 exp-3 without inner seta………………………………………………………….…………*Q. enormis*All body somite covered with minute integumental ornaments, P4 exp-3 with one inner seta…………………..……………….…*Q. sominer*

## 5. Conclusions

In this study, we reported the genus *Quinquelaophonte* from China for the first time, describing a new species of this genus and expanding the distribution of *Q. enormis* from the South Sea of Korea to the North Yellow Sea. We provided a comparison of the most significant morphological characters between the new species and its congeners. The validity of the new species and the monophyly of *Quinquelaophonte* were supported by the molecular phylogenetic analysis results. We also compared the morphological characters of the new record species *Q. enormis* between our samples and the type specimens. Variable setal formulae are not found in the specimens of *Q. enormis* from China. Notably, the setal formula has been considered an important diagnostic characteristic in copepod taxonomy. However, some species belonging to *Quinquelaophonte* are suggested to present a variable setal formula in the second to fourth swimming legs, which brings new challenges to the traditional taxonomy. For this reason, the integrative taxonomy method combining morphological and molecular data is urged for the accurate identification of species belonging to *Quinquelaophonte* in the future.

## Figures and Tables

**Figure 10 biology-14-00271-f010:**
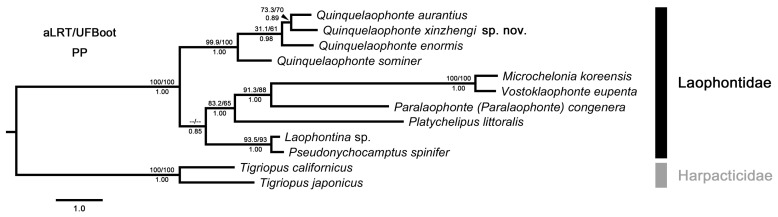
Bayesian inference tree constructed from the concatenated dataset (COI + 18S rRNA genes) partitioned by gene and codon. SH-like approximate likelihood ratio test (aLRT, upper left) values, maximum likelihood ultrafast bootstrap scores (UFBoot, upper right) and Bayesian posterior probabilities (PP, below) are indicated adjacent to each node. Node not recovered by Maximum likelihood analysis is indicated by “--/--”.

**Table 1 biology-14-00271-t001:** Information on the COI and 18S gene sequences used in this study. Sequences in bold were used for phylogenetic tree construction.

Family	Species	Voucher ID	Locality	COI	18S
Laophontidae	*Quinquelaophonte aurantius*	MA73574	New Zealand	**MH444814**	**MH444815**
	*Quinquelaophonte enormis*	NIBRIV0000865946	Korea	**MT416602**	**MT410708**
		MBM189290	China	**PV189950**	**PV189457**
	*Quinquelaophonte sominer*	sed81-06	Korea	**OR659904**	**OR656936**
	*Quinquelaophonte xinzhengi* sp. nov.	MBM189284	China	**PV189951**	**PV189458**
		MBM189285	China	**PV189952**	**PV189459**
	*Platychelipus littoralis*	ACC3	Netherlands	**SRR10208648 ***	**SRR10208648 ***
	*Paralaophonte (Paralaophonte) congenera*	LEGO-HAR027	Korea	**KR049011**	**KR048738**
	*Pseudonychocamptus spinifer*	DZMB010	North Sea	**MF077898**	**MF077714**
	*Laophontina* sp.	DZMB026	Mediterranean Sea	**N/A**	**MF077713**
	*Vostoklaophonte eupenta*	C20	Sea of Japan	**N/A**	**MG012753**
	*Microchelonia koreensis*	C15	Sea of Japan	**N/A**	**MG012752**
Harpacticidae	*Tigriopus californicus*		USA	**DQ913891 ****	**AF363306**
	*Tigriopus japonicus*		Korea	**AY959338 ****	**EU054307**

* COI and 18S gene sequences were extracted from the assembly using the NGS data. ** COI gene sequences were extracted from the complete mitochondrial genomes.

**Table 2 biology-14-00271-t002:** Percentage estimates of Kimura’s 2-parameter pair-wise genetic distances of COI (681 bp, below diagonal) and 18S (1621 bp, above diagonal) gene sequences among studied *Quinquelaophonte* species.

	Species	Voucher ID	Collecting Locality	1	2	3	4	5	6
1	*Q. aurantius*	MA73574	New Zealand	–	1.2	1.2	1.2	0.0	0.0
2	*Q. enormis*	NIBRIV0000865946	Korea	22.9	–	0.0	1.2	1.2	1.2
3		MBM189290	China	22.6	1.1	–	1.2	1.2	1.2
4	*Q. sominer*	sed81-06	Korea	24.8	27.8	27.6	–	1.2	1.2
5	*Q. xinzhengi* sp. nov.	MBM189284	China	26.2	27.0	27.0	26.7	–	0.0
6		MBM189285	China	26.2	27.0	27.0	26.7	0.0	–

**Table 3 biology-14-00271-t003:** Alignment information and selected DNA substitution model in this study.

Gene	Partition Delineation	Subset	Subset Partition	Model Selected by ModelFinder and Implemented in IQ-TREE2	Model Selected by ModelFinder and Implemented in MrBayes
18S rRNA	1–1691	18S rRNA + COI_2nd	1	TNe+I+G4	K2P+I+G4
COI_1st codon	1692–2351\3				
COI_2nd codon	1693–2351\3	COI_1st codon	2	GTR+F+I+G4	GTR+F+I+G4
COI_3rd codon	1694–2351\3	COI_3rd codon	3	GTR+F+I+G4	GTR+F+I+G4

**Table 4 biology-14-00271-t004:** Comparison of characters species in the genus *Quinquelaophonte*, amended from Wells’ results in 2007 [38] and Kim, Nam and Lee’s results in 2020 [3].

Species	No. of Seta									♀ Length Ratio			Type of Elements		References
	♀ A1 Segments	A2 Exp	Gnathobase of Md	Syncoxa of Mxp	Enp of Mxp	♀ P2-P4 enp-2	♀ P3-P4 exp-3	♂ P4 exp-2	P5 Exp/Enp ♀ (♂)	Caudal Ramus	P1 enp-1	P1 enp-2 claw/seta	P1 enp-2	♀ P5 with Bulbous Seta	
*Q. xinzhengi* sp. nov.	6	3	1	2	1	3:5:3	7:6	1	6/5 (5)	3.2	5	≈4.3	2 setae +3 spines	no	this study
*Q. sominer*	6	3	0	2	1	3:4–5:2–3	6:6 ^a^	1	6/5 (5)	3	6	≈2.1	2 setae +3 spines	no	[4]
*Q. enormis*	6	3	0	2	1	3:5:3	5:5 ^b^	0 ^b^	6/5 (5)	3.5	6.1	≈3.8	2 setae +3 spines	no	[3]
*Q. aurantius*	6	3	0	2	1	3:5:3	6:6	0–1	6/5 (5)	3.5–3.8	5.5	≈3.5	2 setae +3 spines	no	[13]
*Q. aestuarii*	6	3	2	2	0	3:5:3	6:6	1	6/5 (5)	4	8	≈7.8	2 setae +3 spines	no	[12]
*Q. varians*	6	3	1	1	0	3:5:3	6:6	0	6/5 (5)	4	5	≈3.6	2 setae +3 spines	no	[11]
*Q. prolixasetae*	6	3	0	2	1	3:5:3	6:5	1	6/5 (5)	3	5	<0.5	5 setae	no	[10]
*Q. koreana*	6	2	0	2	1	3:5:4	7:6	1	6/5 (5)	1.1	5	≈4.7	2 setae +3 spines	no	[9]
*Q. bunakensis*	6	3	1	1	1	3:4:3	7:6	1	6/5 (5) ^c^	2.1	4.7	≈2.5	2 setae +3 spines	no	[8]
*Q. candelabrum*	5	2	0	1	1	3:5:3–4	7:6	1	5/5 (5)	2	5.8	≈3.5	2 setae +3 spines	no	[1,4]
*Q. wellsi*	6	3	0	2	0	3:5:3–4	7:6	1	6/5 (5)	2.7	6.3	≈1.0	2 setae +3 spines	no	[7]
*Q. parasigmoides*	6	3	-	-	-	3:6:3	6:6	1	6/5 (5)	2.5–3.0	-	-	2 setae +3 spines	no	[6]
*Q. longifurcata*	6	3	1	1	1	3:5:4	5–6:5	0	6/5 (5)	4	5.1	≈2.1	2 setae +3 spines	no	[14]
*Q*. *capillata*	6	2	-	1	0	4:6:3	7:5	1	5/4 (5)	2.4	6.3	-	2 setae +3 spines	no	[5]
	6	3	0	2	1	3:5:3	6:6	1	6/5 (5)	3–4	-	-	5 setae	no	[39,40]
	6	3	1	1	1	3:5:3	6:6	-	6/5 (5)	3	5.6	≈2.6	2 setae +3 spines	no	[41]
*Q. quinquespinosa*	6	4	1	1	0	3:5:3	7:-	-	6/5 (5)	2.5	4.6	-	2 setae +3 spines	no	[2,3]
	6	2		2	1	3:5:4	7:6	1	6/5 (5)	2	3	≈4.0	2 setae +3 spines	no	[42]
	6	3	1	2	1	3:5:3–4	7:7	1	6/5 (-)	2.3	3.2	≈3.3	2 setae +3 spines	no	[41]
	6	-	-	-	-	3:5:3	7:7	1	6/5 (-)	3	3.86	≈4.0	2 setae +3 spines	no	[43]

^a^ In Kim and Lee’s description, two specimens had 5 setae or spines in the left P3 exp-3. ^b^ In Kim, Nam and Lee’s description, two specimens had 6 setae or spines on one side of P3 exp-3, and 4 setae or spines on one side of P4 exp-3, and three specimens had one seta on one side of P4 exp-2. ^c^ In Mielke’s description, there were 4 setae close to the outer lobe, but in his figure, only 3 setae were close to that lobe.

## Data Availability

The data presented in this study are openly available in NCBI GenBank at https://www.ncbi.nlm.nih.gov/genbank/ (accessed on 25 February 2025).

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
