# Peer review of "First Report of the Genus Quinquelaophonte Wells, Hicks and Coull, 1982 (Copepoda: Harpacticoida: Laophontidae) from China, with Description of a New Species†"

_biology, 2025, doi:10.3390/biology14030271_

Round 1

Reviewer 1 Report

Comments and Suggestions for Authors

Dear authors

Thank you for the systematic work performed with integrative approach. The work is about the first findings of the genus Quinquelaophonte Wells, Hicks & Coull, 1982 (Copepoda: Harpacticoida: Laophontidae) from China. Two species of the genus were found and one of them is described as a new species in the present work.

The advantage of this work is the use of a complex of morphological and genetic methods. As a result of the analyses, the morphological and genetic data turned out to be congruent. As a result, the description of the species looks reasonable and modern and can be published in the journal «Biology».

Nevertheless, there are some questions and comments on the work and text.

Firstly, the nuclear gene 18SrRNA was chosen not quite correctly. It is too conservative for assessing interspecies relationships. However, given that the authors sequenced a region larger than 1600 bp, it should have shown some differences between species. These differences should be described in the same way as it was done for the mitochondrial gene CO1. If there are no differences, then it is unclear why it was included in the analysis (tree). So far, the work looks like conclusions are made only on the basis of the CO1 gene, which is not quite correct.

Secondly, the variability section (Ñ€.14) provides extremely scant information. Please, provide a more detailed description of variability - what features were assessed; which were conservative within what limits, which features were variable, what are the ranges of variability and so on?

Р.15 Material examined. 3♀♀ remove

The first paragraph of the Discussion with the tables is rather Results section. Please move them to Results. I was not able to access Table 4, as its size more than page size.

After the table probably follows a differential diagnosis? Please mark this section.

In my opinion, the conclusions can be supplemented with information about the discovery of a second species of the genus under study, especially since both the title and the abstract mention this. At the same time, the term chaetotaxy is first mentioned in the conclusions of this article. I think, it would have been worth mentioning earlier or even moving this sentence to the discussion.

So I suppose minor revision is enough to solve all these questions

Reviewer 2 Report

Comments and Suggestions for Authors

Present paper suggests a new Quinquelaophonte and a new record from China with clear text description, nice illustration and molecular evidences. Over all performances are more than enough to be published in Biology, except for few points as follows:

1) Some illustrations need to be reconsidered including a) enlargement of Fig.1 C allobasis for exopod, and absexopodal seta, b) Fig. 3 A, Fig. 5C, Fig.7A and Fig. 8 C are needed to be checked again: Especially authors describe that two abexopodal setae on the allobasis of Antenna. However it is very rare case and I guess that it would be one abexopodal seta and spinules, or a spinule. Setae on the mandibular palp also need to check if they are really fused to the palp, and usually they are clearly distinct and separated from the distal area of palp. Usually there is a pore on the anterior surface of endopodal lobe of P5 in the congeners. In the male P6 bearing somite dose not have serrated frill in the distal margin in male. If this is true, it would be the first case and needs to discuss the details. Setae of P5 in the male also usually distinct and separated form the somite in the congeners, and therefore it should be rechecked.

2) There are some slips of pen and need to be corrected. Those are directly pointed in the text.

Comments on the Quality of English Language

I am a native speaker, and there are no serious matter in the English. I only marked few slips of pen in the text.
